# Bibliometric Review of the Literature on Cone Snail Peptide Toxins from 2000 to 2022

**DOI:** 10.3390/md21030154

**Published:** 2023-02-25

**Authors:** Linh T. T. Nguyen, David J. Craik, Quentin Kaas

**Affiliations:** 1Australian Research Council Centre of Excellence for Innovations in Peptide and Protein Science, The University of Queensland, Brisbane, QLD 4072, Australia; 2Institute for Molecular Bioscience, The University of Queensland, Brisbane, QLD 4072, Australia

**Keywords:** conopeptides, conotoxins, medicinal chemistry, research trends, ziconotide

## Abstract

The venom of marine cone snails is mainly composed of peptide toxins called conopeptides, among which conotoxins represent those that are disulfide-rich. Publications on conopeptides frequently state that conopeptides attract considerable interest for their potent and selective activity, but there has been no analysis yet that formally quantifies the popularity of the field. We fill this gap here by providing a bibliometric analysis of the literature on cone snail toxins from 2000 to 2022. Our analysis of 3028 research articles and 393 reviews revealed that research in the conopeptide field is indeed prolific, with an average of 130 research articles per year. The data show that the research is typically carried out collaboratively and worldwide, and that discoveries are truly a community-based effort. An analysis of the keywords provided with each article revealed research trends, their evolution over the studied period, and important milestones. The most employed keywords are related to pharmacology and medicinal chemistry. In 2004, the trend in keywords changed, with the pivotal event of that year being the approval by the FDA of the first peptide toxin drug, ziconotide, a conopeptide, for the treatment of intractable pain. The corresponding research article is among the top ten most cited articles in the conopeptide literature. From the time of that article, medicinal chemistry aiming at engineering conopeptides to treat neuropathic pain ramped up, as seen by an increased focus on topological modifications (e.g., cyclization), electrophysiology, and structural biology.

## 1. Introduction

The origin of conopeptides contributes to their popularity because it involves the fascinating biology of the extraordinary organisms that produce them, the marine cone snails [1,2,3]. These carnivorous snails from the Conus genus live in tropical and subtropical seas and lie in ambush under the sand or in crevices waiting for their prey to pass by. Like their vegetarian cousins, these snails are slow paced. Nevertheless, they can catch fast-moving fish, or indeed slower moving mollusks or worms, by relying on peculiar hunting strategies [2]. They shoot hollow radular teeth filled with poisonous peptides, i.e., the conopeptides, through an elongated organ called the proboscis, and these teeth act as hypodermic needles to inject a deadly cocktail into their victim [2]. Some cone snails alternatively diffuse conopeptides in the water to anaesthetize a fish or a whole school of fish, enabling the snail to come in the open and engulf their unconscious prey in their mouth before stinging them while captive [2,4].

There are approximately 1000 cone snail species, each producing a mostly distinct set of several hundreds of conopeptides, which are diverse in terms of amino acid sequences, peptide folds, and pharmacological properties [5,6]. Most conopeptides are neuroactive and act on ligand-gated or voltage-gated ion channels with both high potency and selectivity; some of them display picomolar activities and most only target a narrow range of channel subtypes or even a single subtype [1]. This selectivity enables the nervous system of the prey to be subdued by over-activating specific pathways to disrupt cognition while simultaneously inhibiting other pathways related to flight [1]. Because of the large homology between ion channels in the animal kingdoms, conopeptides are often active at human targets. Some of them are considered drug leads, and one is an approved drug, as will be detailed in the Section 3 [5,6]. Conopeptides are also employed as molecular probes in neuroscience to study the involvement of certain ion channels in the nervous system [1,7]. This selectivity is contributed by numerous types of post-translational modifications, including C-terminal amidation, prolyl-hydroxylation, bromination of tryptophan, or phosphorylation, and these modifications considerably expand the chemical space [6,8]. The most common modification is the formation of disulfide bonds, which are important for the stability of peptides. More than 30 disulfide bond scaffolds have been identified so far in conopeptides, and most of those that have been structurally characterized adopt distinct folds [5]. Conopeptides are therefore numerous and diverse in terms of sequence, structure, and chemistry. The marine origin of this natural, highly diverse combinatorial library of ultra-potent and selective peptides appeals to the imaginary and has been qualified as “bounty”, “treasure house”, or “pharmaceutical treasure” [9,10,11].

Shortly after the physiological description of the cone snail envenomation apparatus by Kohn et al. in 1960 [12], Whyte and Endean evidenced the activity of the venom on the nervous system in 1962 [13], but it was only 20 years later that conopeptides started to be individually characterized with the pioneering investigations of Baldomero Olivera, who effectively founded the field [14]. In 1981, he published the first complete amino acid sequences of conopeptides that inhibited muscle contraction [15]. In two following seminal publications in 1985 and 1990 in the journal Science, he revealed that conopeptides are a vast natural peptide family displaying an unparalleled level of diversity and breadth of molecular targets [16,17], seeding an interest that has not faded 30–40 years later. Articles on conopeptides frequently state that conopeptides “attract considerable interest” but there is no definitive analysis to quantify that claim. To fill this gap, we have carried out a bibliometric analysis of the literature on conopeptides from the year 2000 to 2022, providing an unbiased estimate of the breadth of activity in the field, also identifying the main contributors in terms of geographic region, institutions, and investigators, and finally analyzing the major trends of the field over that period through an analysis of publication keywords.

## 2. Materials and Methods

### 2.1. Data Collection

The list of publications on cone snail toxins made over the twenty years from 2000 to 2022 was retrieved from Clarivate Analytics Web of Science (WoS; “https://www.webofscience.com (accessed on 13 January 2023)”). A request was made to the Web of Science Core Collection (WoSCC) bibliographic database using the query “TS = (con-ikot-ikot* OR conantokin* OR coninsulin* OR conkunitzin* OR conoCAP* OR conoGAY* OR conoNPY* OR conodipine* OR conohyal* OR conolysin* OR conomap* OR conomarphin* OR conopeptide* OR conophan* OR conophysin* OR conoporin* OR conopressin* OR conofamide* OR conorphin* OR conotoxin* OR contryphan* OR contulakin*)”. A total of 3451 research articles and reviews were downloaded in the “Web of Science Core Collection” format, which is standardized using a set of two-letter tags and could conveniently be analyzed computationally. We also retrieved the description made in NCBI PubMed (“https://pubmed.ncbi.nlm.nih.gov (accessed on 13 January 2023)”) for each article and review when it was possible. The publication data in PubMed were retrieved using the PubMed identifier provided by WoS. We reasoned that WoS displays a larger breadth of research journals, which should provide a more comprehensive snapshot of the literature than PubMed.

### 2.2. Data Post-Processing

The literature data were first curated where necessary to increase data consistency as well as correct mis- and alternative spellings. The corrections were carried out in four steps, which were implemented in three python scripts provided in the Appendix A.

A manual inspection of the data revealed that several articles were wrongly classified as either reviews or original research. The first python script (“01-retype.py”) re-classified the articles as being either review or original research based on the description made in the database NCBI PubMed in the “PT” field, which we found to be reliable. During that analysis, we also discarded the most recent articles for which no publication date was yet provided (ahead of print articles), as identified using the “PY” fields of WoS.

A second script (“02-rekeywords.py”) was used to homogenize the keywords provided by the authors (“DE” field) and by WoS (“ID” field; known as “Keywords Plus^®^”), enabling the downstream analysis of keyword frequencies. The changes include using the plural of each term (e.g., “isoforms” instead of “isoform”), equivalent terms (e.g., “conopeptides” instead of “Conus peptides”), and renaming peptides because of alternative spelling (e.g., “mu-conotoxin kiiia” instead of “kiiia”) or alternative names (e.g., “omega-conotoxin mviia” instead of “ziconotide”). Several non-informative, albeit frequently used keywords were removed, such as “water” or “identification”. The list of the 594 keyword changes is provided in the Appendix A.

A third script (“03-authors_replace.py”) was used to correct for inconsistent author first names, which are in the “AU” and “AF” fields of the WoS format. The corrections were proposed using a semi-automatic pre-analysis, which involves extracting the frequency of names + first names, names with additional first names, or nearly equivalent names (identical first letters of names). Alternative first names result from partial spelling of the first names, for example, “D”, “DJ”, “D J”, “David”, or “David J” were recorded for “David James Craik”. Occasionally, alternative spellings of first names translated from non-Latin scripts were also identified, such as “Maksim E Astashev” and “Maxim E Astashev”. Names were manually checked for co-occurrence with other names and institutions, resulting in a list of 1964 modifications.

The data used for analysis consisted of 3421 publications, which comprised 3028 original research articles and 393 reviews.

### 2.3. Data Analysis

Our methodology is similar to that described by Zhu et al. (2021) [18], which describes a bibliometric analysis of nicotinic acetylcholine receptors. VOSviewer 1.6.18 [19,20] was used to identify the most active authors, their institutions and countries, as well as the most cited articles and co-cited references. Keywords and trends were analyzed with CiteSpace 6.1.R6 (64-bit) Basic [21].

## 3. Results and Discussion

We retrieved information on 3421 peer-reviewed publications in the field of conopeptides in the period from 2000 to 2022, which should be considered as large, and by itself already proves that it is an important research field. Among these articles, 393 are reviews, representing 11% of the research output as well as an average of 18 reviews per year, which suggests that the field is very active and constantly requires an overview of its current trends to catch up with the latest developments. The cumulative number of research articles, shown in Figure 1, indicates a nearly steady growth in the number of articles in the last 10 years, although at a slower rate than the growth in the decade 2000–2009. The average rate in the 2000–2022 period was 132 ± 37 (standard deviation) publications per year. By contrast, toxins from other animal groups attracted less interest than conopeptides during that period, with <80 publications per year for spider or scorpion toxins. Statistics for other animal groups besides cone snails were made using the PubMed database and in the period 2000-2022; for instance, publications on spider and scorpion toxins were retrieved with the search queries “spider AND toxin AND peptide” and “scorpion AND toxin AND peptide”, resulting in 1178 (56/year) and 1563 (74/year) publications, respectively.

### 3.1. Geographic Regions, Institutions, and Authors

Research on conopeptides is international and is carried out at similar levels in several countries and continents, as shown in Table 1. The USA, where conopeptide research initially sprouted, has the largest publication output from 2000 to 2022, with >1100 publications over that period. The second most productive countries are Australia and China, with ~400 publications. Considering that the pool of scientists in Australia is at least an order-of-magnitude lower than in the USA, Australia’s relative scientific output on conopeptide research is per capita larger than that of the USA. Five European countries, including Germany, France, England, Italy, and Spain, are in the top 10 most productive countries, and together account for >700 publications, and would, therefore place Europe in the second position if its constitutive countries were simultaneously considered. The average number of citations per publication for the top 10 countries is between 23 and 44, therefore, also of a similar level. The impact of the publications made on conopeptides is thus similar in the most productive countries.

Research on conopeptides frequently involves collaborations between laboratories from multiple countries, as illustrated in Figure 2. The 10 countries with the largest publication output are well-connected to each other in this dense network of co-authored publications, suggesting that conopeptide research is the result of international collaborative efforts. International societies related to conopeptide research, such as the International Society on Toxinology (https://www.toxinology.org (accessed on 13 January 2023)) or the International Peptide Society (https://peptidesociety.org (accessed on 13 January 2023)), organize regular symposiums that create opportunities to foster such international collaborations.

In terms of publication output per institution, the University of Utah (USA) and the University of Queensland (Australia) are the most productive, with 398 and 241 publications, respectively, as shown in Table 2. By contrast, the number of publications from the other institutions from the top 10 most productive institutions reaches only 50–60 publications. As we will see in the analyses per authors, the University of Utah and the University of Queensland host more research groups with interest in conopeptides than other institutions. It is interesting to note that the University of Utah has no geographical access to seas into which cone snails live, in contrast to The University of Queensland, which borders the Coral Sea or Hainan University, which is in the South China Sea. The analysis of co-authored publications between institutions shown in Figure 3 confirms that all institutions are interconnected, whatever their country, but also that strong ties exist within countries. For example, most Australian institutions form a cluster comprising the University of Queensland, RMIT University, and the University of Wollongong. Several Chinese institutions, including Tongji University, the Chinese Academy of Science, and Beijing Institute of Biotechnology, form a cluster that is more isolated than others. By contrast, Hainan University does not belong to that cluster but is co-published with the University of Utah, the University of Queensland, and the Russian Academy of Science. The University of Utah has among the strongest co-publications with other institutions, such as the University of Colorado, the University of the Philippines, or the private company Cognetix, which is a biotech spin-out from the University of Utah.

The two most active researchers in conopeptide research in the period 2000–2022 are Michael McIntosh and Baldomero Olivera, with 203 and 182 publications, respectively, as shown in Table 3. Baldomero Olivera founded the field in the 1980s [16] and is one of the two most prolific researchers in this field and in this millennium. His research encompasses the discovery of conopeptides, their pharmacological characterization, and biotechnological applications, especially in medicine. Michael McIntosh is a pharmacologist whose research group is located at the University of Utah, and he has co-authored numerous publications with Olivera, and independently, since 1982 [22]. His focus is on the characterization of ion channels and receptors of the human central nervous system as well as toxins acting on them for physiological studies and drug development. The following four most productive authors in the top 20 list, Richard Lewis, David Craik, David Adams, and Paul Alewood, are all Australian and have, or had, their research group at the University of Queensland. David Adams is currently at the University of Wollongong. As shown in the analysis of co-authorship in Figure 4, these four authors form a tight cluster. They have complementary expertise in natural product discovery (Richard Lewis), peptide chemical synthesis (Paul Alewood, David Craik), structural biology (David Craik), and pharmacology (David Adams), thus enabling productive collaborations for conopeptide discovery and characterization. Interestingly, the clusters of authors (identified by different colors) shown in Figure 4 approximately recapitulate research groups or institutions. For example, the pink cluster is constituted of Sulan Luo (Hainan University) and three other members of her group. The brown cluster also identifies the group of Victor Tsetlin at the Moscow Russian Academy of Science, and the darker blue cluster, albeit more loosely defined, mainly comprises European authors.

### 3.2. Research Trends

The most impactful conopeptide publications in 2000–2022 were investigated both in a global context (Table 4) and within the conopeptide field (Table 5). Six out of the ten most globally influential conopeptide research articles are physiological studies in which conopeptides are used as molecular probes of specific ion channels. In half of these most cited research articles, the α-conotoxin MII was used to identify nicotinic acetylcholine receptors that contain the α6 subunit. Two other articles in that top 10 list focus on the clinical development of the only conopeptide that is approved as a drug, ziconotide (Table 4). Ziconotide, also called ω-conotoxin MVIIA or Prialt, is a conotoxin isolated from *Conus magus* with potent inhibitory activity of the N-type voltage-gated calcium channel, and is an analgesic used to treat chronic pain and pain experienced by cancer and AIDS patients. This analysis suggests that the practical applications of conopeptides attract the most interest from the scientific community.

Six of the ten most cited publications within the conopeptide field, which are listed in Table 5, are reviews (two being the most cited publications), which were either published by the group of Baldomero Olivera or by research groups from the University of Queensland. Three out of the four most field-cited conopeptide research articles are not in the list of the most globally cited publications from Table 4, suggesting that the main interests within the field only partly overlap the interest from researchers out of the field. These differing points of interests are in the discovery of novel conopeptides or in the design of conopeptide variants with altered selectivity. One of these most field-cited articles indeed describes ConoServer [8], which is an expert database on conopeptides and provides essential knowledge and bioinformatic resources for the conopeptide discovery. Another highly cited article focuses on the design of conotoxin MII variants with increased selectivity for α3- or α6-containing nAChR subtypes, with the latter being involved in addiction. The third most field-cited research article reports on the discovery of native peptide CVID, which has a similar molecular target to ziconotide but acts with greater selectivity. Unsurprisingly, the only research article that is both in the most globally cited and field-cited publication lists focuses on the clinical development of ziconotide [23]. This article could, therefore, be described as the article that had the most impact. Ziconotide was discovered by Michael McIntosh as he was working in the laboratory of Baldomero Olivera. After successful clinical development, this conopeptide was the first peptide toxin to be approved by the FDA. After that milestone was reached in 2004, the focus of conopeptide research shifted, as highlighted in Figure 5 by a change in keyword bursts.

Before 2004, the conopeptide literature frequently mentioned voltage-gated calcium channels and spinal cords (both keywords), underpinning the efforts that would ultimately lead to ziconotide approval. Several conotoxins that block voltage-gated calcium channels were heavily mentioned, including GVIA (keyword with highest strength) and MVIIC (keyword), which are used as pharmacological tools due to their very selective and potent (~1 nM IC_50_) activity [1,24]. As discussed above, the approval of ziconotide by the FDA in 2004 was a game changer for peptide toxin research. It also created a drive to find better peptide-based alternatives because (i) ziconotide has poor bioavailability and it is required to be delivered intrathecally, and (ii) this drug is linked to a number of undesirable side effects.

After 2004 and until ca. 2010, the most cited keywords in conopeptide publications relate to physiological and pharmacological studies (dorsal root ganglion, brain slice, ion channel, cDNA cloning), indicating that the broader characterization of conopeptide pharmacological activities was taking place. Although voltage-gated calcium channels are the most cited keyword by research articles from the 2000—2022 period (369 articles), three other ion channels targets were also the focus of numerous articles: the nicotinic acetylcholine receptors (keyword in 265 articles), the voltage-gated sodium channel (121 articles), and the voltage-gated potassium channels (77 articles). During the 2004—2010 period, two conopeptides entered clinical development but were dropped: conantokin-G (CGX-1007), which targets the NMDA receptor and was developed to reduce ischaemic damage in stroke [25], and contulakin-G (CGX-1160), which targets the neurotensin receptor and has analgesic properties [26,27]. Conotoxin MrIA (Xen2174), which inhibits the norepinephrine transporter, went into clinical trial for severe cancer pain in 2008 [28] but it was subsequently dropped [29].

From ca. 2010, efforts were made to improve the pharmaceutical properties of conopeptides through peptide engineering, such as backbone cyclization (keyword) [30]. For example, the backbone cyclization of conotoxin Vc1.1 engendered oral activity in an animal model of neuropathic pain [31]. Rather than screening natural peptides, a range of studies focused on rationally designing conopeptide variants by establishing structure–activity relationships (keyword) and studying the three-dimensional structures of the conopeptides and in complex with their target. Most of the three-dimensional structures of conopeptides were determined by nuclear magnetic resonance (keyword) [5,6], and the complexes between conopeptides and molecular targets were principally modelled using homology modelling (keyword) and molecular dynamic simulations (keyword) [32,33,34]. More recently, several experimental structures of complexes between conopeptides and their targets were determined [34]. For example, the complex between ziconotide and voltage-gated calcium channels was recently determined by cryo-electron microscopy [35], and several structures of complexes between conopeptides and the acetylcholine-binding protein, which is a structural surrogate of the nicotinic acetylcholine receptor, were studied by X-ray crystallography [5]. After 2010, neuronal nicotinic acetylcholine receptors (keyword) gained an increase in focus as a pharmaceutical target for conopeptides [7,36]. These receptors are associated with several diseases and conditions, such as addictions, epilepsy, pain, and Alzheimer’s and Parkinson’s diseases [36]. The exquisite selectivity and potency of some conopeptides for certain subtypes of these receptors made them valuable tools in physiological studies that aim to tease out their involvement in these pathologies [7,37]. Through structure–activity relationship (keyword) studies, a range of conopeptide variants that showed greater selectivity and potency were created, generating valuable new probes and potential new drug leads [38]. For example, RgIA4 is a promising analgesic drug candidate to treat chemotherapy-induced neuropathic pain. It was designed by the substitution of 5 out of the 13 positions (38%) of *Conus regius* conotoxin RgIA [39,40].

Simultaneously as the medicinal chemistry developments took place, the discovery of native conopeptides was accelerated by advances in complementary transcriptomics and proteomics techniques (“mass spectrometry” burst from 2012), rapidly increasing our knowledge in conopeptide natural diversity. These omics studies provided clues on how cone snails generate a massive diversity of toxins, in the range of 1000–2000 per species, from a limited number of genes [41,42,43]. This rapidly growing pool of sequence information was also used to suggest variants for pharmaceutically interesting toxins, therefore supporting conopeptide engineering studies.

Several classification schemes have been developed to describe conopeptides, variously classifying them in terms of their evolutionary relationships (gene superfamilies), activity (pharmaceutical families), disulfide bond numbers and sequence similarity (conopeptides classes), and the pattern of cysteines in their primary sequence (cysteine frameworks are a proxy to describe disulfide bond patterns) [44]. The large amount of sequence and structure data on conopeptides, as well as their evolving nomenclature and classification schemes, prompted the creation of the database ConoServer, the 2012 publication of which is among the top ten field-cited conopeptide publications [8].

Disulfide bonds (keyword) are important because they stabilize peptide three-dimensional structures and are determinants for the overall peptide fold. The importance of the disulfide bonds for conopeptide research is evidenced by the longest citation burst from our analysis, from 2009 until 2022. The structure of proteins and peptides being determinant of their function means that there is in turn a relationship between the disulfide bond pattern and the biological activity of conopeptides [1], albeit conopeptides displaying certain disulfide bond patterns, such as the one forming a cystine knot, is associated with a multiple class of pharmacological targets [44]. There are still large gaps in our knowledge of conopeptides as most of the cysteine frameworks have not yet been structurally studied (17 out of 31 cysteine frameworks) or pharmacologically characterized (19 out of 31 cysteine frameworks).

## 4. Conclusions

The first description of cone snail envenomation was made in 1848 in the scientific report of the HMS Samarang expedition in the East Indies and Southern China [45]. The venom of cone snails, therefore, has generated curiosity and attracted attention for more than 170 years. Our analysis from the years 2000 to 2022 shows that the field of conopeptides is highly active and is carried out worldwide in a spirit of collaboration. Conopeptides indeed do attract “considerable attention”. Since the approval of ziconotide, a major focus of conopeptide research has been on pharmaceutical applications [46] and the use of conopeptides as molecular probes in neuroscience. With the regain in interest from pharmaceutical companies for peptides and the attractive combination of selectivity and potency, newly discovered or designed conopeptides are promising candidates to undergo clinical development and perhaps will join the >80 peptides that have been approved by the FDA and EMA [47,48]. In terms of peptide discovery, <10% of the total pool of conopeptide sequences have been discovered, and only ~300 (<1% of the total) conopeptides have been pharmacologically characterized. We expect that the recent developments in machine learning, which have already revolutionized structural biology [49], could be applied to characterize conopeptide structures and similar algorithms developed to predict their activity, thus potentially helping to discover new “pharmaceutical treasures”. 

## Figures and Tables

**Figure 1 marinedrugs-21-00154-f001:**
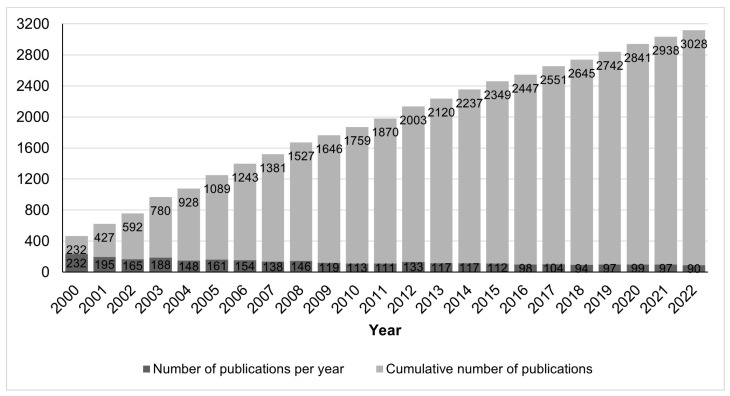
Number of research publications on conopeptides in the period 2000–2022.

**Figure 2 marinedrugs-21-00154-f002:**
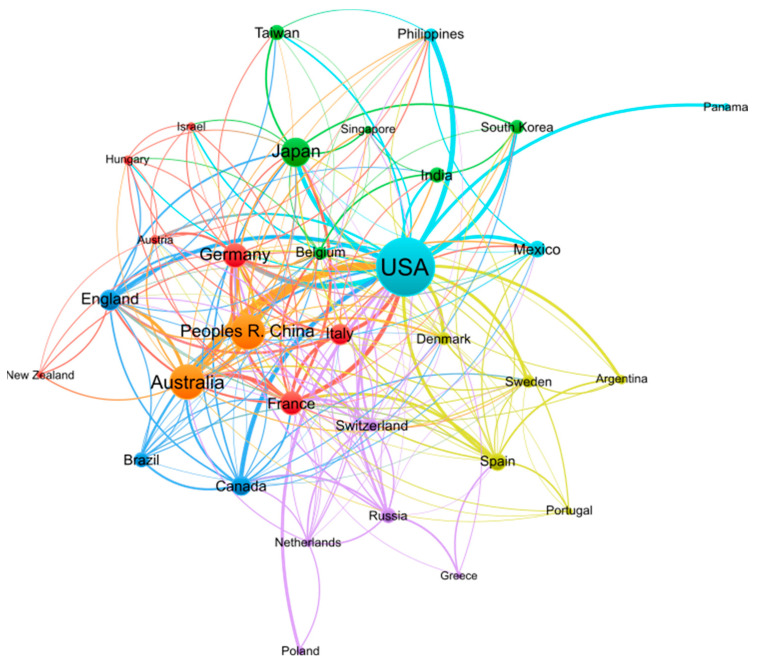
Co-authorship network of countries engaged in conopeptide research in the period 2000–2022. Figure generated by VOSviewer 1.6.18 software with parameters as follows: Type of analysis—Co-authorship; Unit of analysis—Countries; Counting method—Fractional counting; Minimum number of documents of a country—10; the size of the item label in the visualization was determined by the number of documents; the layout was set with Attraction = −2 and Repulsion = −1. All other parameters were set as default values. A total of 32 out of 75 countries meeting the thresholds are shown in the figure.

**Figure 3 marinedrugs-21-00154-f003:**
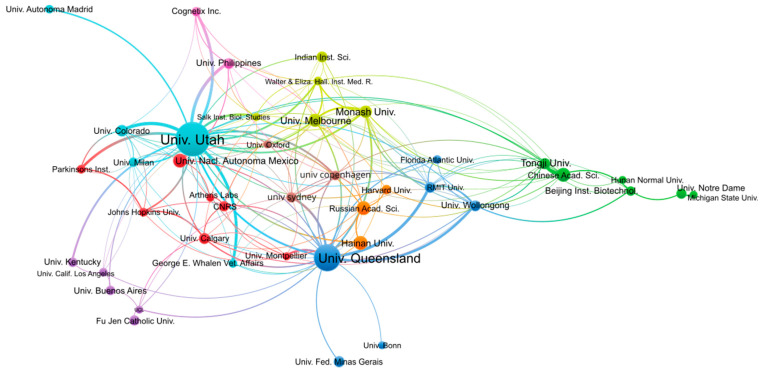
Co-authorship network of institutions engaged in conopeptide research in the period 2000–2022. Figure was generated using VOSviewer 1.6.18 with the following parameters: Type of analysis = Co-authorship; Unit of analysis = Organizations; Counting method = Fractional; Minimum number of documents of an organization = 20; the size of the item label in the visualization was determined by the number of publications; the layout was set with Attraction = 2 and Repulsion = 0. All other parameters were kept with default values. A total of 43 out of the 1950 organizations met the thresholds. The largest set of connected items of the 42 items is shown in the figure.

**Figure 4 marinedrugs-21-00154-f004:**
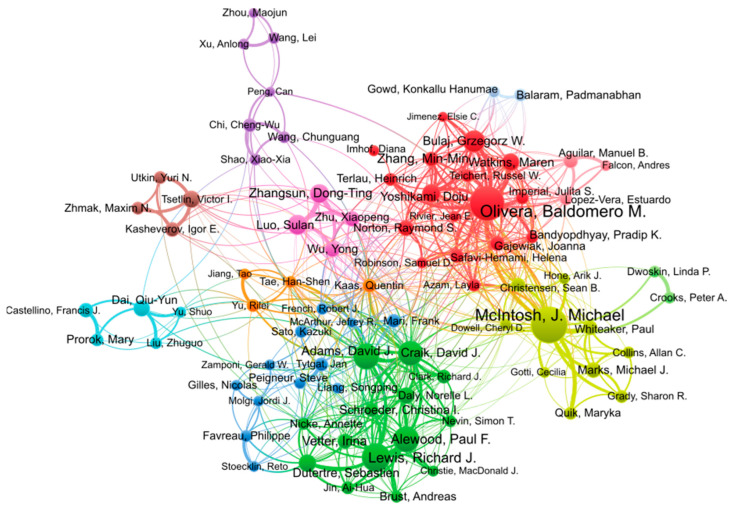
Co-authorship network of authors contributing to conopeptide research in the period 2000–2022. Figure generated using VOSviewer 1.6.18 with the following parameters: Type of analysis = Co-authorship; Unit of analysis = Authors; Counting method = Fractional; Minimum number of documents of an author = 15; the size of the item label in the visualization was determined by the number of publications; the layout was set with Attraction = 2, Repulsion = −2. All other parameters were kept with default values. Of the 9392 authors, 90 authors met the thresholds. The largest set of connected items of 84 items is shown in the figure.

**Figure 5 marinedrugs-21-00154-f005:**
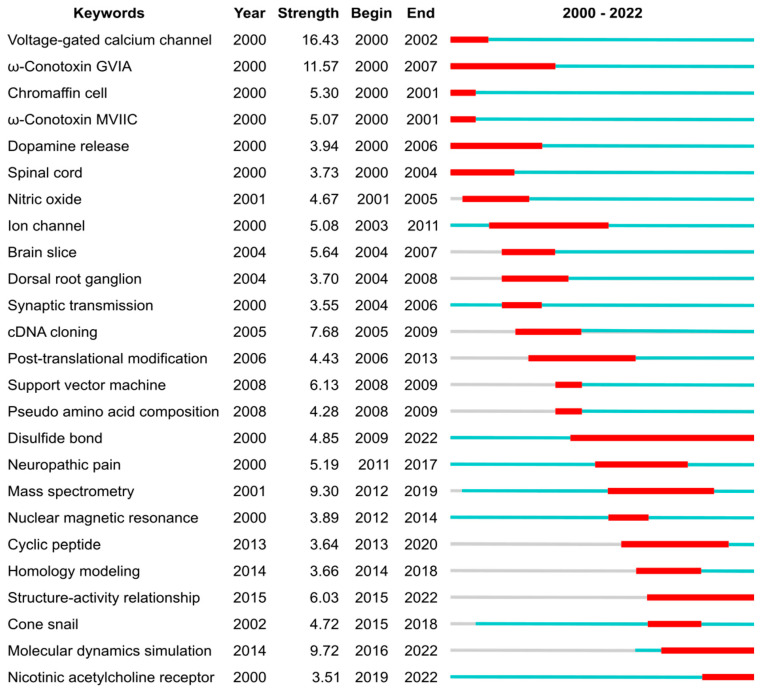
Top 25 keywords with the strongest citation bursts, which represent a frequency surge in a keyword in conopeptide publications during the period 2000–2022. The blue lines depict the periods during when each keyword was used, and the red segments indicate the burst period. The “Year” column is the earliest year when the corresponding keyword had been published. The “Strength” column is the frequency of the corresponding keyword that had appeared over time. The “Begin” and “End” columns are the year of the beginning and end of the burst period, respectively. The analysis was carried out using CiteSpace 6.1.R6 (64-bit) Basic, with parameters as follows: Time Slicing: from 2000 JAN to 2022 DEC, Years Per Slice: 1; Term Source: Title, Abstract and Author Keywords (DE); Node Types: Keyword; Burstness with Minimum Duration: 1.

**Table 1 marinedrugs-21-00154-t001:** Top 10 countries for conopeptide research publications (2000–2022).

Countries/Regions	Documents ^1^	Citations	Average Citation ^2^
USA	1142	42,989	37.6
Australia	412	14,175	34.4
Peoples R. China	395	8350	21.1
Japan	281	6531	23.2
Germany	200	6335	31.7
France	175	7315	41.8
England	147	5925	40.3
Italy	142	6223	43.8
Canada	120	4724	39.4
Spain	91	2165	23.8

^1^ Data were analyzed using VOSviewer 1.6.18 with “Type of analysis” = “Co-authorship” and “Unit of analysis” = “Countries”. ^2^ Average citation was calculated by dividing the number of citations by the number of documents.

**Table 2 marinedrugs-21-00154-t002:** Top 10 institutions for conopeptide research publications (2000–2022).

Organizations	Documents ^1^	Citations	Average Citation ^2^
University of Utah	398	16670	41.9
University of Queensland	241	9369	38.9
Chinese Academy of Sciences	62	1270	20.5
Russian Academy of Sciences	61	1487	24.4
Hainan University	60	788	13.1
Universidad Nacional Autonoma de Mexico	59	926	15.7
University of Melbourne	54	2098	38.9
Monash University	47	1262	26.9
University of Colorado	42	3312	78.9
Royal Melbourne Institute of Technology	40	1206	30.2

^1^ Data were analyzed using VOSviewer 1.6.18 with “Type of analysis” = “Co-authorship” and “Unit of analysis” = “Countries”. ^2^ Average citation was calculated by dividing the number of citations by the number of documents.

**Table 3 marinedrugs-21-00154-t003:** Top 20 active authors in conopeptide research (2000–2022).

Authors	Documents ^1^	Citations	Average Citation ^2^
Mcintosh, J. Michael	203	10,328	50.9
Olivera, Baldomero M.	182	7287	40.0
Lewis, Richard J.	125	4877	39.0
Adams, David J.	96	4373	45.6
Alewood, Paul F.	95	4173	43.9
Craik, David J.	94	4691	49.9
Bulaj, Grzegorz W.	70	2799	40.0
Luo, Sulan	60	788	13.1
Zhangsun, Dong-Ting	54	698	12.9
Yoshikami, Doju	49	2490	50.8
Watkins, Maren	44	1939	44.1
Dutertre, Sebastien	43	1881	43.7
Tsetlin, Victor I.	43	1119	26.0
Dai, Qiu-Yun	42	481	11.5
Daly, Norelle L.	38	1698	44.7
Gomez, Marcus V.	38	845	22.2
Norton, Raymond S.	37	1542	41.7
Zhu, XiaoPeng	37	626	16.9
Marks, Michael J.	34	2844	83.6
Wu, Yong	34	605	17.8

^1^ Data were analyzed using VOSviewer 1.6.18 with “Type of analysis” = “Co-authorship” and “Unit of analysis” = “Authors”. ^2^ Average citation was calculated by dividing the number of citations by the number of documents.

**Table 4 marinedrugs-21-00154-t004:** Ten most cited research articles in the conopeptide field published in 2000–2022.

Research Article ^1^	Citations ^2^	Comment
Hansen, S.B.; Sulzenbacher, G.; Huxford, T.; Marchot, P.; Taylor, P.; Bourne, Y. Structures of Aplysia AChBP Complexes with Nicotinic Agonists and Antagonists Reveal Distinctive Binding Interfaces and Conformations. *EMBO J.* **2005**, 24, 3635–3646.	555	Experimental structure of α-conotoxin ImI in complex with the acetylcholine binding protein.
Klink, R.; de Kerchove d’Exaerde, A.; Zoli, M.; Changeux, J.P. Molecular and Physiological Diversity of Nicotinic Acetylcholine Receptors in the Midbrain Dopaminergic Nuclei. *J. Neurosci.* **2001**, 21, 1452–1463.	553	Physiological studies of nicotinic acetylcholine receptors that use α-conotoxin MII as a molecular probe.
Miljanich, G.P. Ziconotide: Neuronal Calcium Channel Blocker for Treating Severe Chronic Pain. *Curr. Med. Chem.* **2004**, 11, 3029–3040.	434	Clinical development of the first conopeptide-based drug, ziconotide
Champtiaux, N.; Gotti, C.; Cordero-Erausquin, M.; David, D.J.; Przybylski, C.; Léna, C.; Clementi, F.; Moretti, M.; Rossi, F.M.; Le Novère, N.; et al. Subunit Composition of Functional Nicotinic Receptors in Dopaminergic Neurons Investigated with Knock-out Mice. *J. Neurosci.* **2003**, 23, 7820–7829.	407	Physiological studies of nicotinic acetylcholine receptors that use α-conotoxin MII as a molecular probe.
Staats, P.S.; Yearwood, T.; Charapata, S.G.; Presley, R.W.; Wallace, M.S.; Byas-Smith, M.; Fisher, R.; Bryce, D.A.; Mangieri, E.A.; Luther, R.R.; et al. Intrathecal Ziconotide in the Treatment of Refractory Pain in Patients with Cancer or AIDS: A Randomized Controlled Trial. *JAMA* **2004**, 291, 63–70.	399	Clinical trial of the analgesic activity of ziconotide in cancer and AIDS patients.
Salminen, O.; Murphy, K.L.; McIntosh, J.M.; Drago, J.; Marks, M.J.; Collins, A.C.; Grady, S.R. Subunit Composition and Pharmacology of Two Classes of Striatal Presynaptic Nicotinic Acetylcholine Receptors Mediating Dopamine Release in Mice. *Mol. Pharmacol.* **2004**, 65, 1526–1535.	342	Physiological studies of nicotinic acetylcholine receptors that use α-conotoxin MII as a molecular probe.
Zoli, M.; Moretti, M.; Zanardi, A.; McIntosh, J.M.; Clementi, F.; Gotti, C. Identification of the Nicotinic Receptor Subtypes Expressed on Dopaminergic Terminals in the Rat Striatum. *J. Neurosci.* **2002**, 22, 8785–8789.	329	Physiological studies of nicotinic acetylcholine receptors that use α-conotoxin MII as a molecular probe.
Wolf, J.A.; Stys, P.K.; Lusardi, T.; Meaney, D.; Smith, D.H. Traumatic Axonal Injury Induces Calcium Influx Modulated by Tetrodotoxin-Sensitive Sodium Channels. *J. Neurosci.* **2001**, 21, 1923–1930.	299	Physiological studies of voltage-gated calcium channels that use ω-conotoxin MVIIC as a molecular probe.
Champtiaux, N.; Han, Z.-Y.; Bessis, A.; Rossi, F.M.; Zoli, M.; Marubio, L.; McIntosh, J.M.; Changeux, J.-P. Distribution and Pharmacology of α6-Containing Nicotinic Acetylcholine Receptors Analyzed with Mutant Mice. *J. Neurosci.* **2002**, 22, 1208–1217.	291	Physiological studies of nicotinic acetylcholine receptors that use α-conotoxin MII as a molecular probe.
Celie, P.H.N.; Kasheverov, I.E.; Mordvintsev, D.Y.; Hogg, R.C.; van Nierop, P.; van Elk, R.; van Rossum-Fikkert, S.E.; Zhmak, M.N.; Bertrand, D.; Tsetlin, V.; et al. Crystal Structure of Nicotinic Acetylcholine Receptor Homolog AChBP in Complex with an α-Conotoxin PnIA Variant. *Nat. Struct. Mol. Biol.* **2005**, 12, 582–588.	289	Experimental structure of α-conotoxin PnIA in complex with the acetylcholine binding protein.

^1^ Ten most globally cited of the 3028 conopeptide research articles. Data were analyzed using VOSviewer 1.6.18 with “Type of analysis” = “Citation” and “Unit of analysis” = “Documents”. Only research articles from 2000–2022 were considered for this analysis. ^2^ Global number of citations provided by WoS on 13 January 2023.

**Table 5 marinedrugs-21-00154-t005:** The ten 2000–2022 conopeptide publications that were the most cited by the 3421 conopeptide publications in 2000–2022.

Publication ^1^	2000–2022 Citations ^2^	Total Link Strength	Comment
Terlau, H.; Olivera, B.M. Conus Venoms: A Rich Source of Novel Ion Channel-Targeted Peptides. *Physiol. Rev.* **2004**, 84, 41–68.	522	478	Review on conopeptides and their pharmacological activities.
Lewis, R.J.; Dutertre, S.; Vetter, I.; Christie, M.J. Conus Venom Peptide Pharmacology. *Pharmacol. Rev.* **2012**, 64, 259–298.	212	201	Review on conopeptide pharmacological activities.
Miljanich, G.P. Ziconotide: Neuronal Calcium Channel Blocker for Treating Severe Chronic Pain. *Curr. Med. Chem.* **2004**, 11, 3029–3040.	189	179	Clinical development of the first conopeptide-based drug, ziconotide.
Kaas, Q.; Yu, R.; Jin, A.-H.; Dutertre, S.; Craik, D.J. ConoServer: Updated Content, Knowledge, and Discovery Tools in the Conopeptide Database. *Nucleic Acids Res.* **2012**, 40, D325–330.	173	165	ConoServer is an expert database on conopeptides.
Olivera, B.M.; Cruz, L.J. Conotoxins, in Retrospect. *Toxicon* **2001**, 39, 7–14.	151	146	Review on conopeptide early discoveries.
McIntosh, J.M.; Azam, L.; Staheli, S.; Dowell, C.; Lindstrom, J.M.; Kuryatov, A.; Garrett, J.E.; Marks, M.J.; Whiteaker, P. Analogs of α-Conotoxin MII Are Selective for α6-Containing Nicotinic Acetylcholine Receptors. *Mol. Pharmacol.* **2004**, 65, 944–952.	148	142	Development of molecular probes based on α-conotoxin MII selective for α6- or α3-containing nicotinic acetylcholine receptors.
Lewis, R.J.; Nielsen, K.J.; Craik, D.J.; Loughnan, M.L.; Adams, D.A.; Sharpe, I.A.; Luchian, T.; Adams, D.J.; Bond, T.; Thomas, L.; et al. Novel ω-Conotoxins from *Conus catus* Discriminate among Neuronal Calcium Channel Subtypes. *J. Biol. Chem.* **2000**, 275, 35335–35344.	146	129	Discovery of conopeptide CVID, which displays similar activity to the conopeptide-based drug ziconotide but with higher selectivity.
Olivera, B.M. Conus Peptides: Biodiversity-Based Discovery and Exogenomics. *J. Biol. Chem*. **2006**, 281, 31173–31177.	144	140	Perspective on phylogeny-guided discovery of conopeptides.
Akondi, K.B.; Muttenthaler, M.; Dutertre, S.; Kaas, Q.; Craik, D.J.; Lewis, R.J.; Alewood, P.F. Discovery, Synthesis, and Structure-Activity Relationships of Conotoxins. *Chem. Rev.* **2014**, 114, 5815–5847.	143	130	General review on conopeptides.
Lewis, R.J.; Garcia, M.L. Therapeutic Potential of Venom Peptides. *Nat. Rev. Drug Discov.* **2003**, 2, 790–802.	143	135	Review on the potential therapeutic applications of conopeptides.

^1^ Data were analyzed using VOSviewer 1.6.18 with “Type of analysis” = “Co-citation” and “Unit of analysis” = “Cited references”. Both reviews and research articles from 2000–2022 were considered for this analysis. ^2^ The number of citations by the 3421 publications identified using WoS as made by the conopeptide field.

## Data Availability

The code and data used in this study are provided in the Appendix A.

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
