# Peer review of "Bibliometric Review of the Literature on Cone Snail Peptide Toxins from 2000 to 2022"

_marinedrugs, 2023, doi:10.3390/md21030154_

Round 1
Reviewer 1 Report
This is a nice review by Linh T. T. Nguyen et al. This manuscript reviews the literature on cone snail peptide toxins from 2000 to 2022, and presents the current state of research in this area. A bibliometric approach was used to analyze 3,028 research articles and 393 reviews on conopeptides from Clarivate Analytics Web of Science. The study revealed research trends, their evolution, and important milestones in this field.
Comments:
1. Figure 1 shows the cumulated number of publications within the study period. Authors can also provide the exact number of articles published per year.
2. On line 395, the authors mention patents, and it would be perfect if this article could add an analysis of the applications and grants of patents in this field. If there is no relevant introduction, please delete these words
3. About the names of species, please use italicized format. For example, Conus regius on line 349.
4. In the discussion section, authors should discuss future trends in the field of conopeptides research.
Author Response
Comment 1. Figure 1 shows the cumulated number of publications within the study period. Authors can also provide the exact number of articles published per year.
Response. We have added the information about the number articles published each year to Figure 1 and have added a description on lines 147-151.
Comment 2. On line 395, the authors mention patents, and it would be perfect if this article could add an analysis of the applications and grants of patents in this field. If there is no relevant introduction, please delete these words
Response. We have decided not to include the information on patents in our review, which primarily focuses on peer reviewed publications. The word “patent” is not mentioned in our manuscript, and on line 395 the reviewer might have been confused by the words “cysteine pattern”.
Comment 3. About the names of species, please use italicized format. For example, Conus regius on line 349.
Response. All the species names in Latin are now italicized in the revised manuscript.
Comment 4. In the discussion section, authors should discuss future trends in the field of conopeptides research.
Response. Our opinions on the future trends in the fields were already described in the Conclusion section, which follows the Discussion section. We speculated that following the general trends around peptides, toxins will gain an increased focus from the pharmaceutical industry and that deep learning techniques will help to characterize each toxin molecular activities. In the Discussion we also identified upcoming new drug candidates, such as RgIA4, as well as identified knowledge gaps that could be the subject of new research, such as conopeptides displaying poorly characterized cysteine frameworks (lines 393 to 401).
Reviewer 2 Report
In this review the authors provide an exhaustive bibliometric analysis of the published literature on cone snail toxins over the past 20 years, identifying the major countries, institutions and contributors in this field and analyzing the evolution of the different research trends in conotoxin studies.
For example, this review highlights the collaborative approach to conotoxin studies involving multiple countries and the major role of the Universities of Utah and Queensland in this field. In terms of the major research trends in conopeptide studies, this analysis reveals that cone snail peptides have been exploited primarily as pharmacological tools, molecular probes or therapeutic agents.
This review is very clear, well written and highlights the growing interest of researchers in studying and exploiting the cone snail peptide library.
I have only two suggestions :
Even if the analysis of the strongest citation keywords is informative, a bibliometric analysis ranking the main molecular targets (nAChRs, NaV, CaV, GPCRs…) of conotoxins reported in literature could be relevant and added in this review.
Finally, even if it does not exactly fall within the scope of this review, a bibliometric analysis of patents related to the development of conopeptides would also be of great interest.
I therefore recommend the publication of this review in the Marine Drugs special issue on conotoxins.
Author Response
Comment 1. Even if the analysis of the strongest citation keywords is informative, a bibliometric analysis ranking the main molecular targets (nAChRs, NaV, CaV, GPCRs…) of conotoxins reported in literature could be relevant and added in this review.
Response. We have added information about the molecular targets that are the most cited as keywords on page 15 lines 343—347: the voltage-gated calcium channels (369 articles), the nicotinic acetylcholine receptors (265 articles), the voltage-gated sodium channels(121 articles), and the voltage-gated potassium channels (77 articles).
Comment 2. Finally, even if it does not exactly fall within the scope of this review, a bibliometric analysis of patents related to the development of conopeptides would also be of great interest.
Response. We thank the reviewer for this suggestion, but two recently-published reviews already cover the patent literature on conotoxins, and we think that we could not add on these reviews.
Sanchez-Campos N, Bernaldez-Sarabia J, Licea-Navarro AF. Conotoxin Patenting Trends in Academia and Industry. Marine Drugs. 2022; 20(8):531. https://doi.org/10.3390/md20080531.
Durek T, Craik DJ. Therapeutic conotoxins: a US patent literature survey. Expert Opin Ther Pat. 2015;25(10):1159-1173. doi:10.1517/13543776.2015.1054095
Reviewer 3 Report
The manuscript by Nguyen et al. describes a systematic analysis of the literature on conotoxin research, and depicts an overall picture of this field, in terms of both the research work having been done and the researchers who did it. This analysis would be very informative for people in this field and out of this field yet.
This manuscript could be further improved by taking into account the following aspects:
1. The authors state that this field “is very active” with a steady growth, shown by 132 publications per year. It would be more convincing if the overall data of another field is shown as a comparison. The superhot field such as gene-editing is probably not suitable for this comparison, but a classical field could be considered.
2. The field of conotoxin research is kind of unique in the dependence of the wild resources of cone snails that inhabit in only some but all the ocean areas. Therefore, the resource availability is a key factor for the research work that could be done. The correlation, if any, between the distribution of the wild resources and the localization of the researchers/institutions in the field could be included in the manuscript.
3. The manuscript extracted the change of the research in this field by following the trend of keywords, and discussed the future development of this field, which is a nice point. It would be nicer to discuss also the major challenges or obstacles of this field, such as why only Ziconotide goes to the clinical application while a few others all failed, what kind of results will be the breakthrough in this field, etc.
Author Response
Comment 1. The authors state that this field “is very active” with a steady growth, shown by 132 publications per year. It would be more convincing if the overall data of another field is shown as a comparison. The superhot field such as gene-editing is probably not suitable for this comparison, but a classical field could be considered.
Response. It is difficult to compare fields that have very different scope, and we feel that the number of publications by itself already demonstrate the active status of the field. We nevertheless added an analysis in lines 151 to 153 (and footnote) that shows that toxins from other animal groups were less studied over the same period, e.g. 56 and 72 publications per year for spider and scorpion toxins, respectively.
Comment 2. The field of conotoxin research is kind of unique in the dependence of the wild resources of cone snails that inhabit in only some but all the ocean areas. Therefore, the resource availability is a key factor for the research work that could be done. The correlation, if any, between the distribution of the wild resources and the localization of the researchers/institutions in the field could be included in the manuscript.
Response. This is an interesting point and we have added a comment on lines 200—202. The most active institution is the University of Utah, which does not have access to the sea. By contrast, institutions of Australia and China have access to seas where cone snails live.
Comment 3. The manuscript extracted the change of the research in this field by following the trend of keywords, and discussed the future development of this field, which is a nice point. It would be nicer to discuss also the major challenges or obstacles of this field, such as why only Ziconotide goes to the clinical application while a few others all failed, what kind of results will be the breakthrough in this field, etc.
Response. In part “3.2. Research trends”, we presented the trends of conopeptide research in different periods and explained why the trends occurred. We discussed the limitations of conopeptide-based drugs that led to efforts to improve the pharmaceutical properties of conopeptides. These trends are relevant to the major challenges faced by the clinical development of conotoxins as well as of peptides in general, with the most prevalent challenge being the bioavailability. We have highlighted this point on lines 355—358 . The clinical development of conopeptides has hitherto be hindered by the global lower interest of the pharmaceutical companies for peptides, but the growing number of peptides in clinic (discussed in the Conclusion) suggest that this is changing. We also mentioned in the Conclusion that the recent development of machine learning can facilitate the development of conopeptide-based drugs by characterizing conopeptide structures and predicting their activities.
Reviewer 4 Report
Marine Drugs 2023
I am glad that was asked to express my opinion about this unusual review. The authors analyzed 3028 publications in the period 2000-2022 devoted to conotoxins. Although from this list 393 have been already reviews on conotoxins, the present review is not overlapping with them and very clearly describes the present day situation in the field. It mentions that the first indication of the conotoxin action on the nervous system was in 1960 and only 20 years later Professor B. Olivera started the field by determining the first sequence of a conotoxin. The authors presented figures illustrating the most visible labs working in the world on conotoxins and also indicated on these schemes collaborations between different labs all over the world. It is clear that among the leading personalities are Prof. B.M. Olivera and Prof. J.M.McIntosh, but the contributions from Australia are also clearly visible; the important work from China and European countries is also correctly represented.
The authors also clearly presented how focus to one direction of research on conotoxins was shifting with time, especially when ziconotide became the first approved drug inhibiting a certain type of calcium channel and thus providing analgesia to cancer patients. The review also stressed a recent interest to conotoxins attacking the nicotinic acetylcholine receptors involved in the neurodegenerative diseases and inflammation.
Thus, I have no critical comments
Author Response
This reviewer had no comment.